# Trehalose 6-Phosphate/SnRK1 Signaling Participates in Harvesting-Stimulated Rubber Production in the Hevea Tree

**DOI:** 10.3390/plants11212879

**Published:** 2022-10-27

**Authors:** Binhui Zhou, Yongjun Fang, Xiaohu Xiao, Jianghua Yang, Jiyan Qi, Qi Qi, Yujie Fan, Chaorong Tang

**Affiliations:** 1College of Tropical Crops, Hainan University, Haikou 570228, China; 2Rubber Research Institute, Chinese Academy of Tropical Agricultural Sciences, Haikou 571101, China; 3Natural Rubber Cooperative Innovation Center of Hainan Province and Ministry of Education of PRC, Haikou 570228, China; 4Sanya Nanfan Research Institute of Hainan University, Hainan Yazhou Bay Seed Laboratory, Sanya 572025, China; 5College of Biological Sciences and Biotechnology, Beijing Forestry University, Beijing 100083, China

**Keywords:** *Hevea brasiliensis*, laticifers, rubber, harvesting, sucrose, trehalose-6-phosphate (T6P), T6P synthase, SnRK1

## Abstract

Trehalose 6-phosphate (T6P), the intermediate of trehalose biosynthesis and a signaling molecule, affects crop yield via targeting sucrose allocation and utilization. As there have been no reports of T6P signaling affecting secondary metabolism in a crop plant, the rubber tree *Hevea brasiliensis* serves as an ideal model in this regard. Sucrose metabolism critically influences the productivity of natural rubber, a secondary metabolite of industrial importance. Here, we report on the characterization of the T6P synthase (TPS) gene family and the T6P/SNF1-related protein kinase1 (T6P/SnRK1) signaling components in *Hevea* laticifers under tapping (rubber harvesting), an agronomic manipulation that itself stimulates rubber production. A total of fourteen *TPS* genes were identified, among which a class II *TPS* gene, *HbTPS5*, seemed to have evolved with a function specialized in laticifers. T6P and trehalose increased when the trees were tapped, this being consistent with the observed enhanced activities of TPS and T6P phosphatase (TPP) and expression of an active TPS-encoding gene, *HbTPS1*. On the other hand, SnRK1 activities decreased, suggesting the inhibition of elevated T6P on SnRK1. Expression profiles of the SnRK1 marker genes coincided with elevated T6P and depressed SnRK1. Interestingly, *HbTPS5* expression decreased significantly with the onset of tapping, suggesting a regulatory function in the T6P pathway associated with latex production in laticifers. In brief, transcriptional, enzymatic, and metabolic evidence supports the participation of T6P/SnRK1 signaling in rubber formation, thus providing a possible avenue to increasing the yield of a valuable secondary metabolite by targeting T6P in specific cells.

## 1. Introduction

Among living organisms, plants are unique for synthesizing the nonreducing disaccharides sucrose and trehalose. In green algae and primitive plants, including a small number of desiccation-tolerant resurrection plants, trehalose is abundant (in the millimolar range) and functions in osmoregulation and stress protection, as well as carbon storage and transport [1]. However, in most higher plants, trehalose occurs only in the pico- to nanomolar range, with many of the functions in the lower plants taken over by sucrose. Trehalose metabolism is now recognized to play a wide-ranging role in the life of higher plants, despite this carbohydrate occurring in only trace amounts [2]. Trehalose biosynthesis is a two-step pathway catalyzed by trehalose 6-phosphate (T6P) synthase (TPS) that produces an intermediate T6P, followed by the action of T6P phosphatase (TPP) that dephosphorylates the T6P into trehalose [3]. While in the past trehalose was thought to play only trivial roles in higher plants [1], its function is being reviewed following the identification and characterization of multiple *TPS* and *TPP* genes in *Arabidopsis thaliana* [4] and several other plants, such as poplar, rice, and cotton [5,6]. Genetic and transgenic studies have revealed that modifications in trehalose metabolism, especially in its intermediate T6P, have significant effects on plant metabolism, growth, development, and stress response [2,7,8,9,10].

The roles of T6P in regulating starch degradation, sucrose metabolism and allocation in relation to crop yield and stress resilience are gaining recognition [8,11,12,13,14]. In the past several years, significant yield improvement in cereal crops has been realized by directly modifying T6P levels in planta, i.e., in maize through transgenic depression of T6P in ear spikelet tissue (phloem and companion cells) [15], in wheat through increasing T6P in kernels by chemical intervention [16], and in rice through over-expressing the NAC transcription factor OsNAC23 that simultaneously elevates Tre6P and represses trehalose levels [17]. The mechanistic basis of T6P signaling has been investigated in plants, mainly in Arabidopsis. A close correlation between T6P and sucrose is observed both in wild-type and in transgenic Arabidopsis with changed expressions for *TPS* or *TPP* genes [18,19]. T6P seems to act as an indicator of sucrose status, and the T6P:sucrose ratio is critical for regulating sucrose levels within a range conducive to plant growth and development [18]. However, more and more findings reveal that the role of T6P in signaling sucrose availability has a strong tissue and developmental context [8,15,20,21]. Moreover, the sucrose–Tre6P relationship can be disrupted in some complementation Arabidopsis lines for the embryo-lethal tps1-1 null mutant [19]. The signaling effects of T6P function mainly via its inhibition of SnRK1, a SNF1-related kinase sensing energy and carbon availability in all eukaryotic organisms [10,22,23,24]. A model involving the regulation of SnRK1 activity, starch biosynthesis, and sucrose movement was recently proposed to address the underlying mechanisms of yield improvement in cereal crops due to modified T6P in reproductive tissues [8].

Our current knowledge of trehalose metabolism and its role in plants is based mainly on Arabidopsis and a few cereal crops [10], and only recently did relevant studies begin to emerge beyond these plants, e.g., in sugarcane [25,26]. It is, therefore, of interest to elucidate how trehalose metabolism and its associated genes (enzymes) and metabolites (T6P and trehalose) function in different cell types and in other plant species such as the commercial rubber tree *Hevea brasiliensis*. Hevea is widely planted in tropical and sub-tropical Asia and Africa for its production of natural rubber (*cis*-1,4-polyisoprene), an important industrial raw material, and for timber (rubberwood) [27]. Hevea possesses special features that make it a uniquely useful model for conducting such studies. Its laticifer cells are coenocytic, joined end to end. Hence, when the rubber tree is tapped, several tens to a few hundred milliliters of fluid cytoplasm (‘latex’) are expelled from the laticifers and harvested together with natural rubber that makes up about one-third of the latex volume, with the remaining latex content mainly being water [28]. Tapping imposes a stimulatory effect on latex yield, where the regulation of sucrose metabolism in the laticifers is vital for the stimulation of latex regeneration and rubber productivity [29,30,31]. This stimulus is especially conspicuous in previously untapped (‘virgin’) rubber trees. Taking cognizance of the close correlation between sucrose and T6P levels [18], changes in *TPS*-related expression, enzyme activity, and metabolite levels would be useful indicators as to how their functions are inter-related, and how they affect rubber production. Besides losing rubber, the hydrocarbon in the latex when the rubber tree is tapped, a large amount of water is lost from the laticifer system at the same time. No other cell in the plant kingdom suffers a comparable sudden deficit of cellular water content as that experienced in the Hevea laticifer as a result of tapping. A possible role here can, therefore, be contemplated for T6P and trehalose, which are known for their involvement in relief from water stress [15,32]. Under water stress, the slight increase in trehalose levels in drought-tolerant wheat and cotton varieties coincides with higher *TPS* expression [33,34].

The main objectives of this study were: (1) to obtain an overview of the Hevea *TPS* gene family, especially its participation in trehalose metabolism in laticifers; and (2) to investigate the possible involvement of trehalose metabolism, especially T6P/SnRK1 signaling, in water relations and in tapping-stimulated rubber production.

## 2. Results

### 2.1. Profile of the Hevea TPS Gene Family

A total of fourteen *TPS* genes, namely *HbTPS1* to *14* (Appendix A), were identified from the Hevea genome [35]. Except for *HbTPS3* and *4*, full-length cDNAs of the other twelve *HbTPS* genes were successfully isolated by RT-PCR (Appendix A). Specific genomic fragments of *HbTPS3* and *4* were successfully amplified by using genomic DNA as a template (Appendix A), indicating that they are genuine entities in the Hevea genome. All HbTPS proteins shared similar domain organization with a TPS domain (Pfam: Glyco_transf_20) and a TPP domain (Pfam: Trehalose_PPase) in tandem, as revealed by searching the Pfam database. According to their genomic structures (Figure 1A), the fourteen *HbTPS* genes were divided into two distinct classes: four (*HbTPS1* to *4*) in class I and ten (*HbTPS5* to *14*) in class II, having seventeen and three exons, respectively. Phylogenetic analysis that covered 48 plant TPS proteins and one *E. coli* TPS (otsA) was consistent with this division, where the fourteen HbTPS proteins were clustered into the same two classes together with their rice, Arabidopsis and Populus homologs (Figure 1B). The HbTPS proteins had much higher pairwise amino acid (aa) sequence identities within the same classes (73.1–93.3% in class I; 57.7–95.7% in class II) than across the two classes (26.4–30.5%) (Appendix A). Nucleotide (nt) sequence comparisons of the *HbTPS* coding sequences revealed a similar pattern (Appendix A).

Expression patterns of the fourteen *HbTPS* genes in seven Hevea tissues, i.e., latex, mature leaf, trunk bark, mature seed, feeder root, male flower, and female flower, were determined by RNA-Seq analysis (Figure 2A). Of the four class I *HbTPS* genes, *HbTPS1* and *2* showed substantial expressions in most tissues, with the former being much higher than the latter in all cases. In contrast, *HbTPS3* and *4* were either not expressed, or only at very low levels in the seven tissues explored, presumably accounting for our previous failure to clone their cDNAs. The ten class II *HbTPS* genes exhibited distinct patterns of tissue expression, suggesting their differing functions.

In the latex (laticiferous cytoplasm), *HbTPS5* transcripts were much more abundant than those of the other *HbTPS*s, as revealed by both RNA-Seq and qPCR analyses (Figure 2; Appendix A). Immunolocalization analysis further revealed the dominant localization of HbTPS5 proteins in laticifers (Figure 3). As shown in Figure 3A–C, the laticifers were stained dark brown in sections of young Hevea stems and were exclusively localized in the phloem region, near cambial cells. Red fluorescence of anti-HbTPS5 antiserum labeling (Figure 3J) was detected in the granular cytoplasm (latex) of laticifers, overlapping (Figure 3K) with the green fluorescence of anti-REF antiserum labeling (Figure 3I), the REF being known as a laticifer-marker protein [30,36]. In contrast, the pre-immune control for REF and HbTPS5 did not show any fluorescent signal (Figure 3E,F).

To investigate the enzymatic capacity of the HbTPS proteins, yeast complementation assays were conducted using the *tps1Δ* mutant, which grows normally on galactose but not on glucose [37], and its growth on glucose could be restored by the expression of an active TPS enzyme [38]. As shown in Figure 4A, all *tps1Δ* yeast strains transformed with pDR196 expression constructs of various *HbTPSs* grew well on galactose, but only the two class I *HbTPS* isoforms (*HbTPS1* and *2*) could complement the growth defect on glucose, demonstrating them as active TPS enzymes. In addition, the yeast mutant expressing *HbTPS1* or *2* grew much better than the control mutant harboring the empty pDR196 vector on the galactose medium supplemented with 1 mol/L NaCl (Figure 4B), indicating an endowed salinity tolerance.

### 2.2. Characterization of T6P/SnRK1 Signaling Components in Laticifers during the Tapping Treatment

The process of rubber harvesting, i.e., tapping, produces a conspicuous stimulatory effect on latex production in previously untapped (‘virgin’) Hevea trees [39]. The latex outflow increases significantly with each successive tapping until the yield stabilizes at ~10-fold that of the initial tapping after five to seven tappings [29,40]. The crucial roles of sucrose metabolism control in rubber biosynthesis and latex regeneration [41,42] and the presumed signaling roles of T6P in the regulation of sucrose allocation and utilization [8,18] prompted us to investigate whether T6P signaling participates in tapping-stimulated rubber production. As shown in Figure 5A, T6P increased progressively with tapping, reaching 15.3 µM in the latex at the seventh tapping with more than a two-fold rise compared to the first tapping. Trehalose contents increased simultaneously (Figure 5B), with a change of 5.5-fold over the seven consecutive tappings examined. In contrast, sucrose levels fell by 3.7-fold over the same period (Figure 5C). The decreasing sucrose levels are consistent with our previous observations in ‘virgin’ Hevea trees under three different tapping frequencies, i.e., daily tapping, third-day tapping and fifth-day tapping [40]. The decrease in sucrose levels corresponded to tapping-enhanced expression and activity of an alkaline/neutral invertase, HbNIN2, the enzyme responsible for sucrose catabolism in latex [30]. The increase in T6P and trehalose levels might be ascribed to the enhanced activities of the synthesis enzymes, TPS and TPP, by the tapping treatment (Figure 5D–E). Transcripts of *HbTPS1*, the more abundant isoform of the two active TPS genes identified in Hevea (Figure 2A and Figure 4A; Appendix A), were bolstered significantly by the tapping treatment (Figure 5I), corroborating the tapping-stimulated TPS enzyme activity (Figure 5D). However, the expressions of *HbTPS2*, the less abundant active TPS isoform (Figure 2A and Figure 4A; Appendix A), and *HbTPS5*, the latex-predominant *TPS* gene (Figure 2; Appendix A), were both depressed by the tapping treatment (Figure 5I).

One important mechanism in which T6P plays a role is the regulation of carbohydrate utilization through the inhibition of the feast–famine protein kinase, SnRK1 [43]. To explore whether this mechanism manifested itself in Hevea latex, SnRK1 activities were measured in the cytoplasmic serum of latex (C-serum, with endogenous T6P retained) using the AMARA peptide as the substrate. As shown in Figure 5F, SnRK1 activity decreased gradually with the tapping treatment, to 78% of its value by the seventh tapping compared to the initial tapping, contrasting with a gradual rise in T6P levels in the C-serum over the tapping treatment (Figure 5A). When the C-serum was ultra-filtrated to remove endogenous T6P, SnRK1 activity increased by 31% over the seven consecutive tappings examined (Figure 5G). These results together suggest that T6P poses an inhibitory effect on SnRK1 activity in laticifers. To obtain direct evidence, exogenous T6P was added into the ultra-filtered C-serum, and SnRK1 activities were examined. As shown in Figure 5H, SnRK1 activity decreased progressively with increasing T6P added, showing an inhibition of 7% at 5 μM, 16% at 20 μM, and 25% at 200 μM compared with the T6P-free control. Interestingly, transcripts of two *HbSnRK1* genes identified in the Hevea genome [35](Appendix A), *HbSnRK1-1* and *1-2*, decreased as a result of the tapping treatment (Figure 5J), indicating that the regulation of SnRK1 activity in latex occurred at both transcriptional and enzymatic levels.

### 2.3. Analysis of SnRK1 Marker Genes Expression in Laticifers during Tapping

In Arabidopsis, 600 genes are determined as reliable SnRK1 marker genes, and are grouped into two categories: induced (278) and repressed (322) [44]. To examine the functioning of in vivo SnRK1 activity in Hevea laticifers, candidate SnRK1 marker genes were explored by BLAST searching the Hevea genome [35] using corresponding Arabidopsis homologs as queries. A total of 852 SnRK1-induced and 756 SnRK1-repressed Hevea orthologs were identified. Of these genes, the top 200 most abundantly expressed in Hevea latex for each category (Appendix A) were selected, and the average of the normalized expression of each category was examined. As shown in Figure 6A, expression patterns of SnRK1-induced and SnRK1-repressed marker genes exhibited distinct patterns in latex during the tapping treatment, and corresponded to changes in T6P level and SnRK1 activity (Figure 5A,F). The qPCR procedure was further used to determine the expression in Hevea latex of the homologs corresponding to the nine SnRK1 marker genes normally up- (ASN1, bGAL, AKINb, TPS8, TPS10) or down-regulated (UDPGDH, MDH, bZIP11, TPS5) by SnRK1 as reported in Arabidopsis [12]. Of the nine Hevea homologs of candidate SnRK1 marker genes (Appendix A), *TPS5* was excluded from qPCR analysis due to its very low expression in latex, as shown by the RNA-Seq data, and the other eight genes listed represented the most abundant isoforms of corresponding genes expressed in the latex. As shown in Figure 6B, two of the three SnRK1-repressed marker genes, *MDH* and *UDPGDH*, were up-regulated by the tapping treatment. Meanwhile, four of the five SnRK1-induced marker genes, *ASN1*, *AKINb*, *TPS8* and *TPS10*, were down-regulated by the tapping treatment. In brief, six of the eight SnRK1 marker genes examined showed expression patterns in Hevea latex coincident with changes in T6P level and SnRK1 activity (Figure 5A,F), indicating the functioning of T6P/SnRK1 signaling in latex.

## 3. Discussion

### 3.1. The Hevea TPS Gene Family Is Evolutionarily Conserved

A degree of evolutionary conservation appears to be maintained across the *TPS* gene family in higher plants. Among those that have been studied, they have generally similar gene numbers, genomic structure, and most of the ancestral genes are retained. Except for the tetraploid cotton [6], the hexaploid bread wheat and the paleopolyploid soybean [8], which contain more than 20 (22 to 25) *TPS* genes, the six other sequenced plant species (Arabidopsis, rice, poplar and Hevea) explored here (Figure 1B; Appendix A) and previously [5,8] have a dozen or so *TPS* genes: ten in *Brachypodium distachyon*, eleven in Arabidopsis and rice, twelve in Populus, thirteen in maize, and fourteen in Hevea. The *TPS* genes are clustered into two major groups: class I and class II. The genomic structures of *TPS* genes are highly conserved across different plant species to have sixteen introns within the protein-coding regions of most class I genes and two introns within most class II genes [5,6] (Figure 1A). Seven ancestral *TPS* genes have been proposed in the monocot–dicot common ancestor: two (B1 and B2) in class I and five (A1 to A5) in class II [5]. Interestingly, the Arabidopsis genome retains all of the ancestral genes despite being the smallest genome among the higher plants sequenced to date. In contrast, the Populus and rice genomes have lost B2, while the Hevea genome has lost B1 and A5 [5] (Figure 1B). Second, the class I TPS proteins in higher plants conserve most of the amino acid (aa) residues that interact with the substrate (glucose-6-phosphate and UDPG) of the active *E. coli* TPS enzyme (otsA), and tend to be active TPS enzymes. For example, 12 of the 15 aa residues in otsA are conserved in the four Hevea class I TPS proteins (HbTPS1 to 4) (Appendix A), though these HbTPS proteins are distantly related to the otsA protein (Figure 1B). The only three discrepancies are Ile^156^, Glu^186^ and Phe^340^, which are replaced by Thr, His and Leu, respectively, in the four HbTPS proteins. The class I TPS proteins of Populus (PtTPS1 and 2) and rice (OsTPS1) have the same discrepancies (data not shown), but one Arabidopsis homolog (AtTSP4) has an additional difference in Arg^10^, which is replaced by a Ser [38]. In contrast, only a few of these aa residues are conserved in the class II TPS proteins. The conservation status of the active aa is consistent with the functionality of the TPS proteins. In Arabidopsis, all the class I *TPS*, except for *AtTPS3*, which is not expressed and is predicted to be a pseudogene, encode catalytically active TPS proteins, as shown by a yeast complementation assay [38]. Similarly, the sole rice class I TPS protein (OsTPS1) [44] and the two Hevea class I proteins (HbTPS1 and 2) (Figure 4A) are also found to be active TPS enzymes. The other two Hevea class I *TPS* genes (*HbTPS3* and *4*) were not tested in the yeast complementation assay due to the inability to clone their full-length cDNAs. In contrast, no class II *TPS* genes examined to date [38,45,46] encode active TPS enzymes.

### 3.2. HbTPS5 Is Evolved with a Potentially Special Function in Laticifers

In regularly tapped rubber trees, the rubber-producing laticifers represent a strong sucrose sink and are a highly specialized apparatus for rubber biosynthesis [39,42]. It might hence be predicted that some related gene families could have evolved isoforms that function specifically in laticifers. In support of this proposition, *HbTPS5* was found in this study not only to be the predominant isoform expressed in Hevea latex, but its expression in latex was the highest when the expression of all the *HbTPS* genes in all the tissues was compared (Figure 2). As the closest homolog to *HbTPS5*, *HbTPS11* (95.8% identical to *HbTPS5* at the aa level, Appendix A), was expressed at a much lower level, although the two shared rather similar expressions in the other tissues examined (Figure 2A; Appendix A). After BLAST searching the Hevea genome [35], the *HbTPS5* and *HbTPS11* loci were found to be located on the same scaffold (Scaffold50, 2.91-Mb-long), with a spacing of only 9.0 kb. The above data, aided by the duplication–degeneration–complementation (DDC) model [47] and the phylogenetic tree of HbTPS proteins (Figure 1B), allowed us to propose an evolutionary path for *HbTPS5* as follows: first, a segmental duplication of the *HbTPS10* gene region produced a *HbTPS11* ancestral gene; then, a tandem duplication of the *HbTPS11* ancestor gave rise to the *HbTPS5* and *11* gene pair; finally, the further evolution of the *HbTPS5* gene took place to produce a latex-predominant isoform. A number of genes involved in rubber biosynthesis and latex production were also found to be expressed mainly in latex. For example, two *cis*-prenyltransferase genes (*HRT1* and *HRT2*) [48] and four rubber elongation factor/small rubber particle protein (REF/SRPP) genes (*RFE1*, *REF3*, *REF7*, and *SRPP1*) [35] were expressed in Hevea latex. Similarly, three *cis*-prenyltransferase genes (*TbCPT1* to *3*) [49,50], four SRPP genes (*TbSRPP1*, *3* to *5*) [51], one REF protein gene (*TbREF*) [52], and one rubber transferase activator gene (*TbRTA*) [53] were expressed in the latex of dandelion (*Taraxacum brevicorniculatum*). In conjunction with the previous report on *HbNIN2*, a latex-abundant alkaline/neutral invertase gene that is responsible for sucrose catabolism in rubber production in laticifers [30], the identification of *HbTPS5* expression hints at the crosstalk of sucrose and trehalose metabolism in Hevea rubber biosynthesis.

Currently, it is not clear how the seemingly inactive class II TPS proteins exercise their functions in vivo. In *Saccharomyces cerevisiae*, TPS shows optimal enzyme activity only in association with a large multiprotein complex, and several proteins with regulatory functions have been identified from the complex [54,55]. In rice, a yeast two-hybrid (Y2H) assay revealed universal pairwise interactions occurring between its sole active class I TPS protein (OsTPS1) and two inactive class II TPS proteins (OsTPS5 and 8), and among some of its inactive class II TPS proteins, suggesting a regulatory role of ‘inactive’ class II TPSs in controlling TPS enzyme activities [46]. In Arabidopsis, AtTPS8-10 are repressed by sucrose, and they likely have regulatory functions too [12,19]. However, the Y2H test in the present study (Appendix A) showed no interactions between the two active class I HbTPS proteins (HbTPS1 and 2) and the 10 inactive class II HbTPS proteins, including HbTPS5, the laticifer-predominant isoform. Therefore, it remains to be precisely determined in what ways HbTPS5 participates in T6P synthesis, or directly in T6P signaling in laticifers.

### 3.3. T6P/SnRK1 Signaling Is Involved in Rubber Production Stimulated by Tapping

As rubber biosynthesis occurs in Hevea laticifers with sucrose as the precursor molecule, the regulation of sucrose metabolism is vital to rubber production [42]. In recent years, it has become clear that T6P signals sucrose availability, and is at the center of controlling sucrose allocation and use in various sinks [9,10,15,20,43,56]. Latex harvesting (tapping) stimulates rubber yield conspicuously in previously untapped Hevea trees [29,31,40]. Although the underlying mechanisms are not fully understood, the latex flow-stimulating effect has been ascribed substantially to enhanced sucrose uptake and sucrose catabolism in laticifers [29,30]. Thus, the tapping treatment of ‘virgin’ Hevea trees constitutes an ideal model for studies on T6P biosynthesis and its signaling function in laticifers pertinent to rubber production.

In untapped Hevea trees, the metabolism of laticifers is in a resting state in relation to rubber biosynthesis, and the commencement of tapping acts as a trigger to turn on the metabolic switch [39,57]. In regularly tapped Hevea trees, the laticifers function as an active sink for sucrose, the consumption of which is exploited to replenish the lost latex before the next tapping [42]. The up-regulation by tapping of a sucrose transporter, HbSUT3 [29], and an alkaline/neutral invertase, HbNIN2 [30], facilitates sucrose uptake and catabolism in laticifers. Our recent transcript and protein profiling studies revealed that the activation of sugar transport and metabolism as well as rubber biosynthesis is vital to harvesting-induced latex production [31]. In the present study, the tapping treatment resulted in a gradual rise in T6P levels (Figure 5A), indicating good sucrose availability [12], and the promotion of sucrose utilization in laticifers (Figure 5C). Similar to studies in other plants [43], T6P signaling functions in Hevea laticifers, at least partly through inhibiting the feast–famine protein kinase SnRK1. This is evidenced by the decrease in SnRK1 activity with the tapping treatment (Figure 5F), the inhibition of SnRK1 activity by exogenous T6P (Figure 5H), and the correlation of SnRK1 marker genes expression with changes in SnRK1 activity in latex during the treatment (Figure 6). Transcriptional and enzymatic regulation of T6P biosynthesis (Figure 5D,I) and transcriptional regulation of *SnRK1* genes (Figure 5J) by means of the tapping treatment further corroborate the involvement of T6P/SnRK1 signaling in laticifers. Trehalose levels in the latex (Figure 5B) were comparable or lower than those reported in other plants [18,58,59], but increased markedly when the tree was tapped. Such low levels preclude the role of trehalose as an effective osmolyte in the laticifers, but it might function as a potential signal metabolite [2,9,33,34] implicated in the relief of water-loss stress in laticifers (bearing in mind that water is the major constituent of latex) and, along with T6P, in tapping-stimulated rubber production. A working model is presented to summarize these findings (Figure 7).

In conclusion, our study has not only catalogued the complete TPS gene family in the Hevea tree, but provided transcriptional, enzymatic and metabolic data supporting the participation of the resulting T6P metabolite and its target kinase SnRK1 in harvesting-stimulated latex production. The predominant expression of *HbTPS5* in the rubber-producing laticifers and its apparent regulation by the harvesting treatment underpin the regulatory function of this enzymatically inactive class II TPS gene in the T6P signaling pathway in relation to latex production.

## 4. Materials and Methods

### 4.1. Plant Materials

Mature rubber trees (*Hevea brasiliensis*) used in experiments were of the cultivar Reyan7-33-97 planted at the experimental plantation of the Rubber Research Institute, Chinese Academy of Tropical Agricultural Sciences, Danzhou, Hainan, China. Young stems of Reyan7-33-97 trees were collected from the National Rubber Tree Germplasm Repository, Danzhou, Hainan, China.

The effect of tapping on the expression of relevant genes in the latex was tested on 8-year-old ‘virgin’ (previously untapped) rubber trees. These trees were tapped at a frequency of every three days, and latex samplings were taken at the first, third, fifth and seventh tapping. The rubber tree was tapped by shaving off a sliver of bark about 1.5 mm thick from the surface of a half-spiral tapping cut into the tree trunk. The laticifer plugs that had sealed the cut ends of the laticifers to stop latex exudation in the preceding tapping were thus removed along with the bark shaving; latex flow from the newly unplugged laticifers ensued. To study the expression of *HbTPS* genes in different tissues (latex, mature leaf, trunk bark, mature seed, feeder root, male flower, and female flower), samples were collected from 10-year-old mature rubber trees that had been tapped for the past two years [60]. Young stems of rubber trees with shoots cut back 5 months earlier were used in immunolocalization experiments as described previously [30]. The latex collected for RNA extraction in the tapping treatment was also exploited to determine the enzyme activities of TPS, TPP and SnRK1 and levels of sucrose, trehalose-6-phosphate (T6P) and trehalose.

### 4.2. RNA and DNA Extraction

Genomic DNA was extracted from the leaves of mature rubber trees using the CTAB method [61]. Total RNA was isolated from the latex [62] and other Hevea tissues [29]. RNAs and DNAs were quantified spectrophotometrically at 260 nm and 280 nm by using an ultraviolet-visible spectrophotometer (T6, Beijing Puxi General Instrument Co., Ltd., China) and checked by means of agarose gel electrophoresis [63].

### 4.3. Cloning of cDNA and Genomic DNA

To identify *TPS* and *SnRK1* homologues in *H. brasiliensis*, the *TPS* and *SnRK1* genes of *Populus trichocarpa*, *Arabidopsis thaliana* and *Oryza sativa* were used as query sequences to BLAST search the Hevea transcriptome [64] and genome sequence [34]. The retrieved transcripts and genomic contigs were used to predict the full-length cDNAs of *TPS* and *SnRK1* genes. First-strand cDNA was synthesized using DNase I-treated RNA samples of the different Hevea tissues, and served as a template to amplify the full-length cDNAs of 14 *HbTPS* and 2 *HbSnRK1* genes using gene-specific primer pairs (Appendix A). To verify the presence of *HbTPS3* and *4* in the Hevea genome, specific primer pairs were designed to amplify partial genomic DNAs of respective genes (Appendix A). The PCR products obtained were cloned into the pMD18-T vector (TaKaRa) and sequenced at Invitrogen Company (Shanghai, China).

### 4.4. Expressional Analysis Based on RNA Sequencing

Latex cDNA libraries for the tapping treatment (tappings 1, 3, 5 and 7) were prepared using the Illumina Truseq RNA sample preparation kit following the manufacturer’s instructions, and pair-end (PE) sequencing of the cDNA libraries was completed on the HiSeq2000 system (Illumina, San Diego, CA, USA). PE reads were mapped using Bowtie2 to quantify transcript abundance by RSEM software. The RNA-Seq data of the seven Hevea tissues were as acquired previously [60]. Expression level of each gene was scored as RPKM (Reads Per Kilobase per Million mapped reads).

The entire sets of 600 reliable SnRK1 marker genes as reported in Arabidopsis [44] were downloaded on 23 November 2016 from the TAIR database (https://www.arabidopsis.org/) and used as queries to identify corresponding homologs in Hevea using the BLAST program. Their transcript abundance in the latex was analyzed by RNA-Seq for the four tappings (tappings 1, 3, 5 and 7), and normalized against the first tapping. The expression trend chart was drawn using R scripts.

### 4.5. Quantitative RT-PCR

Quantitative RT-PCR (qPCR) was performed as described previously [29]. The primer pairs used for the *HbTPS* and *HbSnRK1* family genes, and SnRK1 marker genes were as listed in Appendix A. The reaction was performed using the Light Cycler 2.0 system (Roche Diagnostics, Penzberg, Germany) with the SYBR Green premix kit (TaKaRa) according to the manufacturer’s instructions. The efficiency of each primer pair was evaluated and found to be between 1.844 and 1.997. For internal controls, *YLS8* was used in the analysis of gene expression in the latex following the tapping treatment [65].

### 4.6. Antibody Preparation and Immunolocalization

A monoclonal mouse anti-HbTPS5 antibody was developed on 1 July 2015 by AbMart (www.ab-mart.com.cn) using the HbTPS5 fragment of amino acids 363 to 372 (KQHPKWQGRA). Antibodies against the laticifer-abundant protein, rubber elongation factor (REF), were prepared as described by [30]. Immunolocalization of the HbTPS5 proteins was conducted using the young stems from Reyan7-33-97 trees that had been cut back, with REF as the laticifer marker protein. Laboratory procedures were as previously described [30]. Iodine-bromine staining was performed on paraffin-embedded sections to locate laticifers in the young rubber stems, as described by [66].

### 4.7. Yeast Complementation Assay and Growth Response to Salt Stress

Experiments were conducted with the wild-type *Saccharomyces cerevisiae* strain W303-1A (*MAT*a *ade2-1 his3-11*, *15 leu2-3*, *112 trp1-1 ura3-1 can1-100 GAL mal SUC2*) and its *tps1* mutant (W303-1A, *tpsΔ::TRP1*). To obtain the *tps1* mutant, the endogenous *ScTPS1* gene was replaced by the *TRP* marker gene as previously described [67,68]. The transformed yeast cells were screened on synthetic dropout (SD) 2% galactose (Gal) medium lacking tryptophan (Trp) at 30 °C. The *tps1* mutant cells, but not the wild type cells, grew on SD-Trp.

The plasmid pDR196 with the *PMA1* promoter and *URA3* marker was used as an expression vector [69]. Each of the full-length *HbTPS* ORFs was amplified from recombinant pMD18-T vectors harboring the *TPS* cDNA, and then digested and inserted into the pDR196 vector. To enhance the translational efficiency of each *TPS* ORF in yeast, the sequence AAGCTTGTAAAAGAA taken from the 5′-flanking region of *AtSTP1*, which has been verified as being ideal for the yeast translation machinery [70], was added just before the ATG initiation codon in the forward primers used in the PCR amplification. The primer pairs used are shown in Appendix A. As a TPS positive control, the *ScTPS1* gene was isolated by PCR from yeast genomic DNA using gene-specific primers (Appendix A). The recombinant pDR196 plasmids were transformed into the mutant yeast cells using the PEG/LiAc method [71], and then selected on SD-Ura 2% Gal plates. The pDR196 vector was transformed into the mutant yeast as a negative control. The positive transformants were cultured in SD-Ura Gal liquid medium to OD_600_ = 1.0, and then 5 µL of each transformant was dripped on SD-Ura 2% Gal or 2% glucose plates, as previously described [54].

To determine the growth response of the *TPS* transformants to salt stress, the transformants containing pDR196-*HbTPS1*, pDR196-*HbTPS2* or pDR196 were cultured to OD_600_ = 1.0, and the culture diluted to an OD_600_ value of 0.04. Five microliters of each dilution was spotted on SD-Ura 2% Gal plates supplemented with 1mol/L NaCl and cultured at 30 °C for two days before photographic recording.

### 4.8. Yeast Two-Hybrid Analysis

Yeast two-hybrid analysis was conducted using the Matchmaker™ Gold Yeast Two-Hybrid System (Clontech, Mountain View, CA, USA) according to the manufacturer’s instructions. The full-length ORFs of *HbTPS1*, *2*, and *5* were amplified using gene-specific primer pairs (Appendix A), and inserted into pGBKT7. Meanwhile, the full-length ORFs of 12 *HbTPS* genes (*HbTPS1*, *2*, *5* to *14*) were amplified using gene-specific primer pairs (Appendix A) and inserted into pGADT7. One pGBKT7 construct was co-transformed with one pGADT7 into yeast strain Y2H Gold, and the interaction test was performed according to [46].

### 4.9. TPS, TPP and SnRK1 Enzyme Assays

The clear Hevea latex cytoplasmic serum (C-serum), separated from fresh latex by high-speed centrifugation as described previously [30], was used to determine the enzymatic activities of TPS, TPP and SnRK1. To remove low molecular metabolites, C-serum samples were filtered through syringe filter units (0.45 µm) (Millipore, Burlington, MA, USA), followed by centrifugal ultrafiltration with Amicon Ultra-15 (NMWL, 30 kDa) (Millipore). Protein concentration of the samples was determined by the method of Bradford [72].

TPS activity was assayed according to [55] with slight modifications. Briefly, the assay mixture (250 µL) consisted of 50 mM Tris-HCl buffer (Ultra Pure Grade, 99%, Solarbio, pH 8.5), 10 mM MnCl_2_ (Grade A. R., 99%, Sinopharm, Beijing, China), 1 µg heparin salt (Grade I-A, Sigma, St. Louis, MO, USA), plus 5 mM UDPG (≥95%, Sigma) and Glu6P (Reagent grade, ≥98%, Sigma) as substrates and 50 µL of C-serum as the crude enzyme. The assay mixture was incubated at 37 °C for 1 h, following which HCl (Grade A. R., 36.0–38.0%, Sinopharm) was added to a final concentration of 0.5 M and placed in boiling water for 10 min to destroy the residual UDPG. NaOH (Grade A. R., 97%, Sinopharm) was added to the concentration of 1.0 M, and then boiled as above to destroy the residual reducing sugars. Trehalose-6-phosphate (T6P) synthesized in the mixture was not affected by the above treatments, and was determined by the anthrone colorimetric method against standard T6P by using ultraviolet-visible spectrophotometer (T6, Beijing Puxi General Instrument Co., Ltd., Beijing, China). C-serum inactivated by boiling water was used in enzyme blanks. Enzyme activity (U) for TPS was expressed as μmol of T6P synthesized per min under the assay conditions.

TPP activity was measured as described [73] by determining the release of inorganic phosphate (Pi) from T6P using ultraviolet-visible spectrophotometer (T6, Beijing Puxi General Instrument Co., Ltd., Beijing, China). TPP enzyme activity was expressed as μmol of Pi released from T6P per min under the assay conditions.

SnRK1 activity was assayed according to [22] with modifications. The reaction mixture of 25 µL contained 40 mM HEPES-NaOH (Grade A. R., 98%, Sinopharm, pH 7.5), 5 mM MgCl_2_ (Grade A. R., ≥98%, Sinopharm), 200 µM ATP containing 0.2 µCi [γ-^32^P] ATP (GE Healthcare), 200 µM AMARA peptide (AMARAASAAALARRR; AnaSpec), 5 mM dithiothreitol (Grade A. R., ≥95%, Sigma), 1 µM okadaic acid (≥90%, Sigma), 1 mM protease inhibitor cocktail (Sigma-Aldrich P8340), and 3 μL of latex C-serum. After 6 min of incubation at 30 °C, 15 µL was transferred to 4 cm^2^ squares of Immobilon-P Transfer Membrane (0.22 µm PVDF) (Millipore) that had been treated by soaking in methanol for 5 min and kept wet. After air drying, the membrane squares were washed four times each for 1 min with 800 mL of 1% phosphoric acid (Grade A. R., ≥85%, Sinopharm), air dried, and transferred to vials containing a 2 mL scintillation cocktail (OptiPhase Supermix; Perkin Elmer, Waltham, MA, USA). Incorporation of ^32^P was determined by liquid scintillation counting (MicroBeta; Perkin Elmer). The specific activity of SnRK1 was calculated as nmol of inorganic phosphate (Pi) incorporated into AMARA peptide per min per mg of C-serum protein in the reaction mixture.

### 4.10. Determination of Sucrose, T6P and Trehalose Levels

Sucrose content in the latex was measured according to [74]. T6P content was quantified in the C-serum by hydrophilic-interaction liquid chromatography-mass spectrometry (HPLC-PDA-Deca MAX-MSn, Thermo-Finnigan, Silicon Valley, CA, USA) as described [75]. Trehalose content in the C-Serum was determined using the Megazyme trehalose assay kit (K-TREH 11/12, Megazyme, Bray, Ireland) according to the manufacturer’s protocol.

### 4.11. Bioinformatics Analysis

The analyses for the cDNAs, predicted proteins, and gene structures were performed by GENETYX-WIN (version 4.06) and the Softberry software (http://linux1.softberry.com/berry.phtml) (accessed on 23 June 2014). Homology analyses (nucleotide and amino acid) of the fourteen *HbTPS* genes were generated with DNAMAN V6. Phylogenetic relationships were analyzed among the fourteen HbTPS proteins and 34 other previously reported TPS sequences of Populus [76], Arabidopsis [4], and rice [46] using MAGA version 6.0. The neighbor-joining method was used to build the phylogenetic tree with *Escherichia coli* TPS (otsA) used as an outgroup. The protein domains of the HbTPS proteins were searched using the Pfam database (http://pfam.xfam.org/search accessed on 23 June 2014).

### 4.12. Statistical Analysis

Student’s *t*-test was performed using the software embedded in Excel 2007 for comparison of *HbTPS* and *HbSnRK1* expression, TPS, TPP and SnRK1 activities, and sucrose, T6P and trehalose levels under different experimental conditions. Differences were accepted as significant at *p* < 0.05.

## Figures and Tables

**Figure 1 plants-11-02879-f001:**
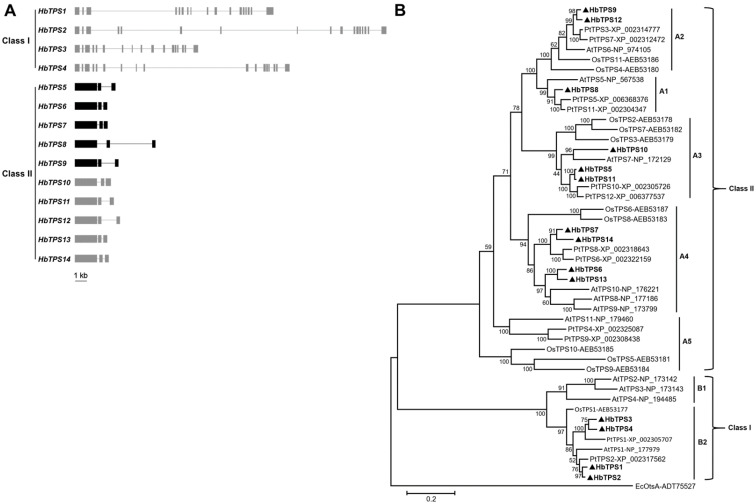
Exon–intron structure of the fourteen *HbTPS* genes (**A**) and their protein phylogeny (**B**). (**A**) Exons are shown by solid boxes and introns by solid lines. The genomic structures represented by gray boxes and lines were predicted using the Hevea genome draft sequence, whereas those in black boxes and lines were further confirmed by PCR cloning and sequencing. (**B**) The tree is constructed by using MEGA version 6.0 with 48 plant TPS proteins that include 11 in Arabidopsis thaliana, 11 in Oryza sativa, 14 in H. brasiliensis, and 12 in Populus trichocarpa, and the *E. coli* otsA as an outgroup. The fourteen HbTPS proteins are highlighted with black triangles.

**Figure 2 plants-11-02879-f002:**
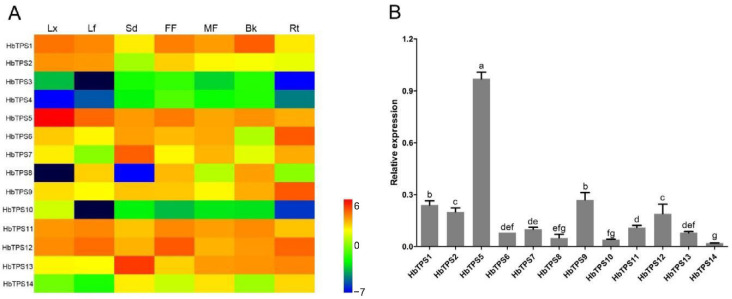
Tissue expression patterns of *HbTPS* genes. (**A**) Expression of 14 *HbTPS* genes in seven Hevea tissues, viz. latex (Lt), leaf (Lf), seed (Sd), female flower (FF), male flower (MF), bark (Bk), and root (Rt). The heat map is drawn using the R package based on the Log 2 reads per kilobase exon per million mapped reads (RPKM) of the RNA-Seq data. (**B**) Relative expression of 12 *HbTPS* genes in Hevea latex by qPCR. Values are means ± standard deviations of three biological replicates. Different letters indicate significant differences across different *HbTPS* genes (Student’s *t*-test was performed using Excel 2019, *p* ≤ 0.05).

**Figure 3 plants-11-02879-f003:**
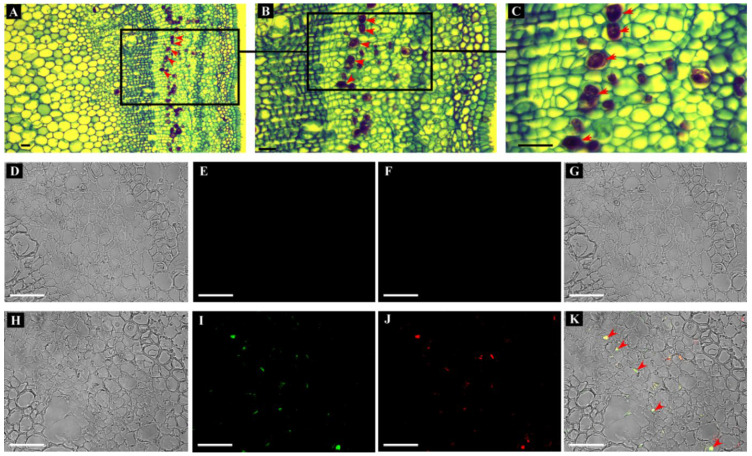
Immunohistochemical colocalization of HbTPS5 and laticifer-specific REF proteins by fluorescence detection. Transverse sections of young rubber stem were subjected to iodine-bromine staining, and the laticifers are visualized with a brown coloration (**A**–**C**; arrow heads). Similar sections were used for immunolocalization analysis (**D**–**K**). (**H**) Bright field image of sections used for immunolocalization in (**I**–**K**). Confocal microscopy images of sections reacted with the REF antibody and a secondary antibody conjugated with FITC (**I**), and the HbTPS5 antibody and a secondary antibody conjugated with DyLight650 (**J**). (**K**) shows a merged image of (**H**–**J**). Pre-immune controls for the proteins of REF and HbTPS5 are shown in (**D**–**G**). Note the co-localization of HbTPS5 antibody and REF antibody in the laticifers demonstrated by a yellow coloration of the merged signals of HbTPS5 (red) and REF (green) (arrowheads). Bars = 20 µm.

**Figure 4 plants-11-02879-f004:**
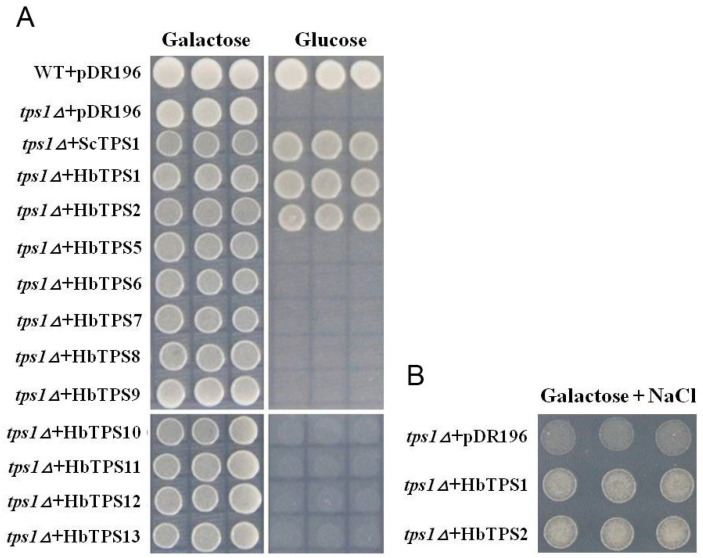
Complementation assay identifies two active *HbTPS* isoforms that confer salt tolerance in yeast. (**A**) Complementation assay of 12 HbTPS proteins in yeast *tps1* mutant grown on SD medium with galactose or glucose. Three types of control strains were included, i.e., wild-type yeast with the empty pDR196 (WT+pDR196), yeast mutant with the *ScTPS1* expression construct (*tps1Δ*+*ScTPS1*), and yeast mutant with the empty pDR196 (*tps1Δ*+pDR196). (**B**) Identical amounts of cells from yeast *tps1* mutants harboring various constructs were spotted on galactose medium supplemented with 1 mol/L NaCl. The yeast cells were grown at 30 °C for 2 days.

**Figure 5 plants-11-02879-f005:**
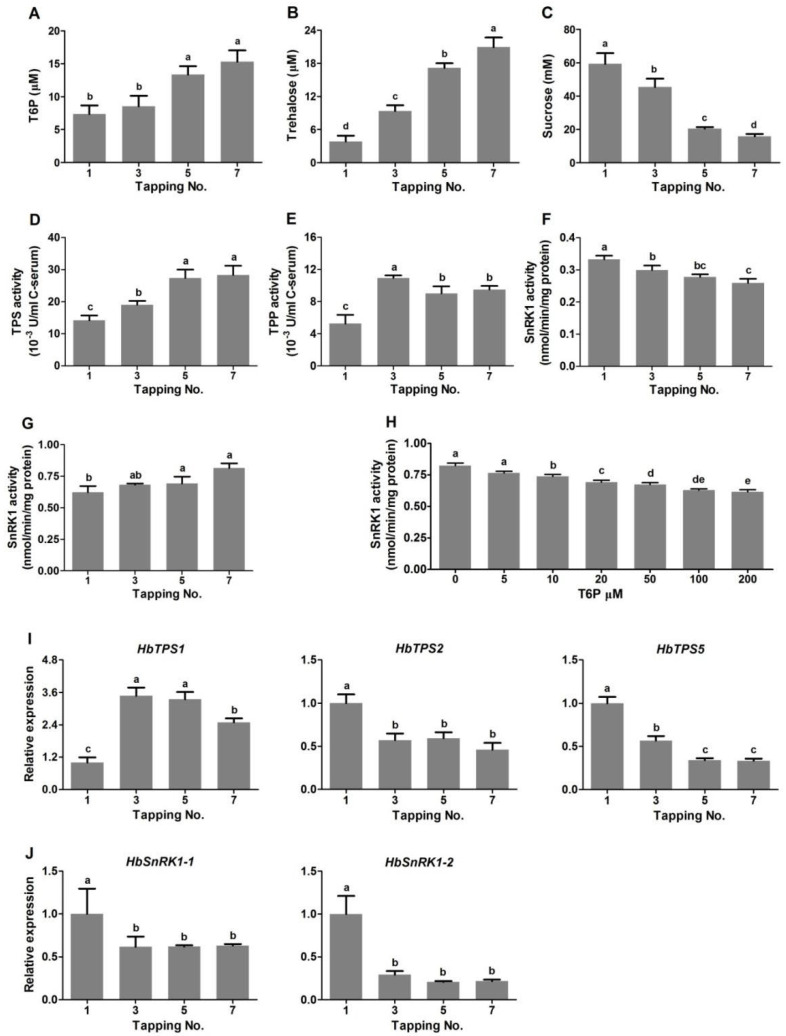
Metabolic, enzymatic, and transcriptional characterization of T6P-SnRK1 signaling components in Hevea latex during tapping. (**A**–**C**) Changes in T6P (**A**), trehalose (**B**), and sucrose (**C**) levels. (**D**–**G**) Changes in enzyme activities of TPS (**D**), TPP (**E**), and SnRK1 (**F**) in C-serum samples. SnRK1 activity in ultra-filtered C-serum (**G**) and dose effect of T6P on its activity (**H**). (**I**), Changes in expression of three *HbTPS* genes, *HbTPS1*, *2* and *5*, by qPCR. (**J**) Changes in expression of two *HbSnRK1* genes, *HbSnRK1-1* and *1-2*, by qPCR. Values are means ± standard deviations of three biological replicates. Different letters indicate significant differences (Student’s *t*-test was performed using the software embedded in Excel 2019, *p* ≤ 0.05).

**Figure 6 plants-11-02879-f006:**
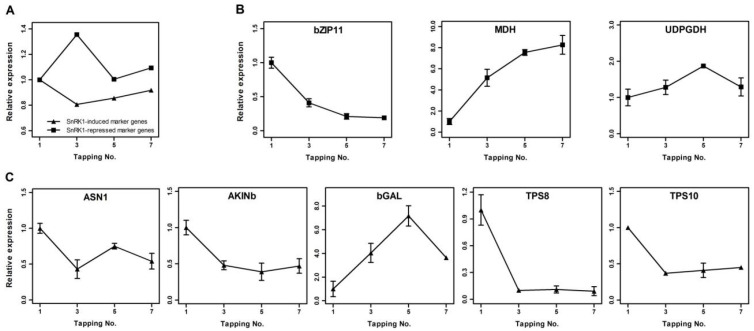
Changes in SnRK1 target gene transcript abundance in Hevea latex during tapping. (**A**) A total of 1032 putative SnRK1 target genes (511 induced and 521 repressed) reported in Arabidopsis (Baena-Gonzalez et al., 2007) were used as queries to identify their corresponding homologs in Hevea using the BLAST program. The expression of each target gene in latex during the tapping was scored in RPKM based on RNA sequencing data generated from the Illumina platform and normalized against the first tapping. Two hundred each of the most abundantly expressed SnRK1-induced and SnRK1-repressed target genes are presented. (**B**) Expression of three putatively down-regulated SnRK1 targets, *bZIP11*, *MDH* and *UDPGDH*, analyzed by qPCR during tapping. (**C**) Expression of five putatively up-regulated SnRK1 targets, *ASN1*, *AKINb*, *bGAL*, *TPS8* and *TPS10*, analyzed by qPCR during tapping. The qPCR data are means ± standard deviations of three independent biological samples, normalized against the first tapping.

**Figure 7 plants-11-02879-f007:**
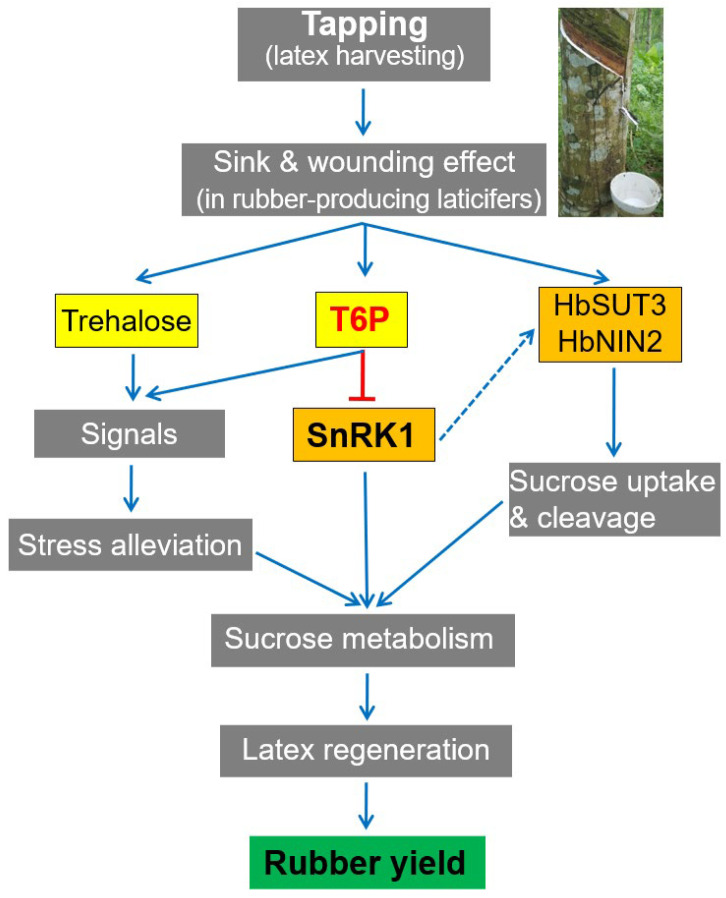
Model for T6P-SnRK1 signaling in harvesting-stimulated rubber production. Tapping incurs sink and wounding effects in rubber-producing laticifers. This increases T6P and trehalose levels and the expression of a sucrose transporter, HbSUT3, as well as an alkaline/neutral invertase, HbNIN2. Elevated T6P functions in at least two ways to enhance sucrose metabolism in laticifers: in activating a ‘feast’ response by inhibiting SnRK1 activity and, together with trehalose, as a signal to relieve water loss stress in a SnRK1-independent manner. The inhibited SnRK1 might also facilitate sucrose uptake and catabolism in laticifers through the induction of HbSUT3 and HbNIN2. The strengthened sucrose metabolism promotes latex regeneration, and thus the rubber yield.

## Data Availability

The data used in this study are available from the corresponding author on submission of a reasonable request.

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
