# Peer review of "Trehalose 6-Phosphate/SnRK1 Signaling Participates in Harvesting-Stimulated Rubber Production in the Hevea Tree"

_plants, 2022, doi:10.3390/plants11212879_

Round 1
Reviewer 1 Report
Paper was well written and in the scope of the journal however some queries should be addressed;
1. The introduction contained sufficient reference but most of the references are old and need to be updated with recent studies from 2021, 21, 22.
2. In the methodology, the complete name of the instrument along with model, make and country should be mentioned throughout the section.
3. Grade of the chemicals, their purity and brand should be mentioned throughout the methodology section.
5. Probability value to check the significance should be ≤ 0.05 not < 0.05.
6. Also mention the name of software with complete detail on which the statistical analysis was performed.
7. A comprehensive conclusion should be the part of the manuscript.
8. Results and discussion section should be improved with the recent literature citations.
9. Some figures need to be more clear.
Reviewer 2 Report
This is a clearly written study that presents new information about the T6P signalling pathway in the rubber crop.
In the abstract. This is not the first study to document secondary metabolism affected by T6P. Oszvald et al. (2018) showed effects on secondary metabolism in maize. doi: 10.1104/pp.17.01673
Neofunctionalised. I am not sure this term is clear. Why not say HbTPS5 may have evolved or been selected in rubber with a function likely associated with latex formation? However, as its expression decreases with tapping and is inversely related with sucrose levels it may perform a regulatory function in the T6P pathway. A number of Arabidopsis TPSs (AtTPS8-10) are repressed by sucrose and they likely have regulatory roles too. I am not sure that decreased expression of HbTPS5 confounds the understanding of trehalose metabolism if the above context is considered. You may want to change use of "confound" e.g. in abstract.
In Discussion, wheat and other crops have TPS and TPP sequences that are known too, first paragraph of discussion. The list is more comprehensive than documented here and you may want to mention this. See Paul et al. 2018 doi: 10.1104/pp.17.01634.
